# In Situ Synthesis of NiFeLDH/A–CBp from Pyrolytic Carbon as High-Performance Oxygen Evolution Reaction Catalyst for Water Splitting and Zinc Hydrometallurgy

**DOI:** 10.3390/ma16113997

**Published:** 2023-05-26

**Authors:** Kai Che, Man Zhao, Yanzhi Sun, Junqing Pan

**Affiliations:** State Key Laboratory of Chemical Resources Engineering, College of Chemistry, Beijing University of Chemical Technology, Beijing 100029, China; chekai121@163.com (K.C.); sunyz@mail.buct.edu.cn (Y.S.)

**Keywords:** OER catalyst, pyrolytic carbon black, water splitting, anodic catalyst, zinc electrowinning

## Abstract

Nickel–iron-layered double hydroxide (NiFeLDH) is one of the promising catalysts for the oxygen evolution reaction (OER) in alkaline electrolytes, but its conductivity limits its large-scale application. The focus of current work is to explore low-cost, conductive substrates for large-scale production and combine them with NiFeLDH to improve its conductivity. In this work, purified and activated pyrolytic carbon black (CBp) is combined with NiFeLDH to form an NiFeLDH/A–CBp catalyst for OER. CBp not only improves the conductivity of the catalyst but also greatly reduces the size of NiFeLDH nanosheets to increase the activated surface area. In addition, ascorbic acid (AA) is introduced to enhance the coupling between NiFeLDH and A–CBp, which can be evidenced by the increase of Fe-O-Ni peak intensity in FTIR measurement. Thus, a lower overvoltage of 227 mV and larger active surface area of 43.26 mF·cm^−2^ are achieved in 1 M KOH solution for NiFeLDH/A–CBp. In addition, NiFeLDH/A–CBp shows good catalytic performance and stability as the anode catalyst for water splitting and Zn electrowinning in alkaline electrolytes. In Zn electrowinning with NiFeLDH/A–CBp, the low cell voltage of 2.08 V at 1000 A·m^−2^ results in lower energy consumption of 1.78 kW h/Kg_Zn_, which is nearly half of the 3.40 kW h/Kg_Zn_ of industrial electrowinning. This work demonstrates the new application of high-value-added CBp in hydrogen production from electrolytic water and zinc hydrometallurgy to realize the recycling of waste carbon resources and reduce the consumption of fossil resources.

## 1. Introduction

The electrochemical oxygen evolution reaction (OER) has important applications in many fields, including water electrolysis, electrochemical synthesis, Zn hydrometallurgy, etc. [1,2]. The OER is a four-electron process with a sluggish kinetic process that requires a high overpotential, largely increasing the energy consumption of the whole electrocatalytic process [3,4]. Therefore, researchers are committed to developing high-performance electrocatalysts to effectively reduce the overpotential of the reaction process and the energy consumption [5,6,7,8]. The high cost and low reserve of precious metal catalysts with high electrochemical performance limit their large-scale application. Hence, exploring new non-noble metal-based catalysts with low cost, high efficiency, and stability is the focus of the current work.

Nickel–iron-layered double hydroxide (NiFeLDH) is one of the promising catalysts for OER in alkaline electrolytes, with a large specific surface area and layered structure facilitating charge exchange. However, the poor conductivity and incomplete exposure of active sites hinder the at-scale application of NiFeLDH. In order to improve the conductivity of NiFeLDH, researchers have loaded NiFeLDH nanosheets onto a conductive substrate. For example, Guofa Dong et al. synthesized LDH nanosheet arrays containing NiFe on carbon fiber cloth (CC) [9]. In another study, Ming Gong et al. synthesized ultra-thin nickel iron layered double hydroxide (NiFe LDH) nanosheets on lightly oxidized multi walled carbon nanotubes (CNTs) [10]. Jiahao Yu et al. in-situ grown 3D mesoporous NiFe layered double hydroxide (LDH) micro clusters with layered structure on foam nickel [11]. In addition, Abolfath Eshghi et al. electrochemically synthesized nickel–iron-layered double hydroxide nanocomposites on graphene/glassy carbon electrodes [12]. The above scholars combined LDH with conductive materials to obtain catalysts that exhibit good OER performance in alkaline solutions. However, the disadvantages of these works are obvious: the high cost or complex synthetic method of the substrate or the weak interaction between NiFeLDH and the substrate. Therefore, developing a conductive substrate with a low-cost, simple synthetic method and strong interaction with NiFeLDH is a promising approach to improving the catalytic performance of NiFeLDH, achieving mass production.

CBp is a product from the pyrolysis of waste tires under anaerobic conditions at 450 °C [13,14],which is mainly composed of different grades of commercial carbon black and inorganic fillers from the rubber manufacturing process, with the ash content being 15–20% [15,16,17]. Its price is only 5–10% of that of commercial CB, and the application is mostly limited to the rubber field [18,19]. In order to improve the added value and broaden the application fields of CBp, researchers have developed its applications in energy storage and catalysis. For example, Lisa Djuandhi et al. demonstrated significant advantages in the application of CBp as an electrode component in lithium-ion battery systems [20]. Shilpa et al. extracted customized activated carbon from CBp as a new anode material for lithium-ion batteries [21]. Ravi Kali converted the rubber tubes of discarded bicycles into value-added low-dimensional carbon materials and used them as negative electrode materials for sodium-ion battery applications [22]. Next, Keqiang Ding et al. prepared honeycomb-shaped carbon particles from CBp as anodes for lithium-ion batteries [23]. Chun Chi Chen et al. also investigated the application of nitrogen-doped CBp as an electrode material for supercapacitors [24]. Another study converted pyrolysis carbon derived from discarded tires into activated carbon for supercapacitor electrodes [25]. Another converted waste tire rubber into high-value-added carbon carriers for electrocatalysis [26]. The above examples from the literature show the potential applications of CBp in energy storage, but the purity of CBp still limits its performance. CBp with high purity and high activity is necessary to further improve performance. Based on the composition characteristics of CBp, our team obtained activated CBp with high purity and high specific surface area through acid–base purification treatment and KOH activation [27,28], which will further promote the application of CBp as a substitute for activated carbon in adsorption, energy storage, and catalysis [21,29,30,31,32]. Hence, CBp is a promising candidate as a conductive substrate of NiFeLDH, due to its large supply and low cost, along with the simple and large-scale processing technology from our group.

Furthermore, with the reduction of high-grade and high-quality sphalerite resources, the shortage of zinc resources has become acute [33]. The recovery of metallic zinc from ZnO ore and waste slag by alkaline solution electrolysis has become the major method of zinc production, with advantages such as high efficiency, low pollution, high yield, and good operability. S. V. Mamyachenkov et al. studied the electrolytic recovery of zinc in alkaline zincate solutions under laboratory conditions, which could provide satisfactory coulomb efficiency (85–95%) and low application energy consumption (2.28–3.20 kW h/KgZn) [34]. Zhao et al. reported a new comprehensive hydrometallurgical process for producing zinc powder in alkaline media, demonstrating its superiority in alkaline electrolysis [35]. S. Gürmen et al. studied the possibility of applying alkaline electrolysis technology to electrolytic zinc, which saves 20% in terms of energy compared to industrial electrolytic zinc [36]. In the alkaline electrolysis process, the anode for the OER significantly impacts the whole zinc electrodeposition process, requiring good electrocatalytic performance, good conductivity, stability, long life, low corrosion resistance, and affordable production costs [37,38,39]. This situation is similar to hydrogen production by water electrolysis. The development of OER catalysts with excellent catalytic performance, low cost, and suitability for mass production is a long-term goal. Combining NiFeLDH with CBp as anodic catalyst for zinc recovery and water electrolysis should be an effective solution.

Based on the above considerations, activated CBp with high purity and specific surface area obtained through acid–base purification and KOH activation was used as a conductive substrate to prepare NiFeLDH/A–CBp through a facile in situ synthesis method. In order to improve the OER catalytic performance of NiFeLDH/A–CBp, ascorbic acid (AA) as a chelating agent was introduced to enhance the coupling between NiFeLDH and A–CBp and the related synergistic effect of NiFe atoms. The optimal catalyst with a large active surface area offered a low overpotential (227 mV) for water splitting in the alkaline electrolyte. In addition, the NiFeLDH/A–CBp catalyst largely reduced the bath voltage to reduce the energy consumption during the electrolytic process. Therefore, NiFeLDH/A–CBp, as a promising catalyst, is expected to enlarge the commercial application of Zn recovery and water electrolysis.

## 2. Materials and Methods

### 2.1. Materials

Pyrolytic carbon black (CBp) was provided by Hunan Qiheng Environmental Protection, China. Urea (CO(NH_2_)_2_, 71334-76-4, 99%), ethyl alcohol (EtOH, 141-78-6, 95%), hydrochloric acid (HCl, 7647-01-0, 36%), and sulfuric acid (H_2_SO_4_, 8014-95-7, 51.9%) were received from Beijing Chemical Factory, China. Ammonium fluoride (NH_4_F,14972-90-8, 99.5%), iron nitrate nine hydrate (Fe (NO_3_)_3_·9H_2_O), nickel nitrate hexahydrate (Ni (NO_3_)_2_·6H_2_O), and ascorbic acid (C_6_H_8_O_6_, 2252244-20-3, 99.7%) were provided by Xilong Chemical Industry, China. Zinc oxide (ZnO, 1314-13-2, 99%) and sodium hydroxide (KOH, 1310-58-3, 85%) were supplied by Beijing Chemical Factory, China. Carbon paper, foam nickel, and foam nickel molybdenum were provided by Cyber.

### 2.2. Synthesis of NiFeLDH/A–CBp

A–CBp was prepared by purification and KOH activation according to the method of a previously published paper [27,28]. The impurities in CBp were leached with acid to obtain purified carbon black, which was denoted by P–CBp. Then, P–CBp was activated with KOH (ω:ω = 1:4) for 90 min at 700 °C under N_2_ atmosphere. The activated product was washed and dried to obtain activated carbon black (A–CBp).

NiFeLDH/A–CBp was prepared by a hydrothermal method. Firstly, mixtures with 50 mg A–CBp, 0.15 mmol Fe (NO_3_)_3_·9H_2_O, and different contents of AA were ultrasonically dispersed in 50 mL water for 30 min. Then, 3.5 mmol NH_4_F, 0.6 mmol Ni(NO_3_)_2_·6H_2_O, and 15 mmol CO(NH_2_)_2_ were added to the above solution, which was stirred for 10 min at 60 °C. The resulting mixture was heated at 120 °C for 12 h. Lastly, the samples were collected, washed, and dried. Here, the obtained catalysts with different AA contents from 0 mmol to 0.4 mmol are named AA–x mmol. Among them, the optimized OER catalyst is marked as NiFeLDH/A–CBp.

### 2.3. Characterization

The following characterization methods were used to examine the structure, morphology, and surface properties of the prepared materials, including SEM, TEM, XRD, FTIR, and XPS. The detailed information of the characterization devices is shown in Appendix A.

### 2.4. Electrochemical Measurement

Electrochemical properties were measured by a three-electrode system. The prepared catalysts were applied as working electrodes, Pt plate as a counter electrode, and Hg/HgO as the reference electrode.

Electrodeposition of zinc and water electrolysis were measured by LANDIAN using a two-electrode system. The prepared catalysts were applied as anode electrodes. Copper sheets and foam nickel molybdenum were the cathode electrodes for zinc electrodeposition and water electrolysis, respectively.

## 3. Results

The ash, accounting for almost 20% of CBp, mainly includes colloid layers, coke from the pyrolysis process of waste tires, and inorganic fillers like SiO_2_, Zn, Ca, etc., that weaken the performance of CBp [40]. Therefore, as Figure 1 shows, it is essential to purify and active CBp by removing impurities with acid and baking with KOH before synthesizing a catalyst. Then, the activated CBp (A–CBp) can be combined with NiFeLDH to form an NiFeLDH/A–CBp catalyst by in situ hydrothermal method.

In Figure 2a, micro-scale NiFeLDH nanosheets with high crystallinity assemble into a big globular flower [41]. When NiFeLDH nanosheets are combined with A–CBp in Figure 2b, their size decreases obviously, and they cannot be uniformly dispersed on the surface of A–CBp, only gathering together. In order to enhance the interaction of NiFeLDH nanosheets with ACBp, ascorbic acid (AA) is introduced. When the AA content is 0.1 mmol (Figure 2c), the layered structure of NiFeLDH nanosheets can uniformly load onto the A–CBp surface to form a layered structure to fully expose active sites and improve conductivity. As the AA content increases to 0.2 mmol in Figure 2d, the interaction between A–CBp and NiFeLDH nanosheets further strengthens, and part of the surface of the NiFeLDH nanosheets is covered by A–CBp to block active sites. When continuing to increase the content of AA, the NiFeLDH nanosheets and A–CBp adhere to each other as shown in Figure 2e,f, which greatly reduces active site exposure. Hence, the proper content of AA (0.1 mmol) is beneficial to charge transfer between NiFeLDH nanosheets and A–CBp [42].

The morphology of the synthesized NiFeLDH/A–CBp was studied through TEM (Figure 3). The introduction of AA enables NiFeLDH to be completely encapsulated on the surface of CBp, ensuring a strong interface connection between the two [42]. The HRTEM images in Figure 3b,d,f,g,h clearly show the lattice of (015) of NiFeLDH [43,44,45,46,47]. The lattice stripe spacing decreases with the increase of AA content, indicating that NiFeLDH tends to be amorphous, mainly due to the enhanced force between Fe and Ni, which is consistent with SEM results [48].

The XRD patterns of NiFeLDH/A–CBp samples with various AA contents are shown in Figure 4a. The results of AA–x mmol are well-matched with the standard pattern (JCPDS card No.38-0715) [42]. The displayed peaks at 11.35°, 22.74°, 33.46°, 34.41°, 38.77°, 45.99°, 59.98°, and 61.25° are ascribed to (003), (006), (101), (012), (015), (018), (110), and (113), respectively [42]. Compared with AA–0 mmol, the intensities of all peaks of NiFeLDH/A–CBp gradually decrease with the increase of AA content. This shows that AA strengthens the interaction between NiFeLDH and A–CBp while weakening slightly the orderliness of NiFeLDH nanosheets grains.

The surface chemical properties of NiFeLDH/A–CBp samples with different AA contents were studied using FTIR spectroscopy, as shown in Figure 4b. The peaks at 3425 and 1618 cm^−1^ belong to the stretching vibrations of hydroxyl groups and C=O, which are related to AA and CBp [49]. In addition, the peaks of NiFeLDH/A–CBp at 1358 and 1492 cm^−1^ are the interaction of NO^−^_3_ interlayer groups [50]. The spectral bands within the range of 500–900 cm^−1^ are related to the metal–oxygen–metal lattice vibrations of NiFeLDH/A–CBp (Fe–O and Ni–O) layer cations. The band appearing at approximately 670 cm^−1^ is attributed to the stretching mode of Fe–O–Ni [51]. In addition, there is a strong peak at 1620 cm^−1^, which is mainly derived from CBp [52]. Among the signals, due to the introduction of AA, the NO_3_ peak intensity weakens, and the peak intensities of Fe–O–Ni and M–O become stronger, indicating a stronger interface connection between NiFeLDH and CBp.

The XPS analysis of NiFeLDH/A−CBp with AA–0 mmol and AA–0.1 mmol were shown in Figure 4c. The binding energies (284.6 eV and 531.9 eV) are C ls and O 1s, respectively [42]. In the synthesis process, there is a small amount of Ni^3+^ and Fe^2+^. For NiFeLDH/A−CBp with AA−0.1 mmol, XPS spectra displays decomposed peaks of Ni 2p at 858.4 eV, 864.2 eV, 875.9 eV, and 880.3 eV, corresponding to Ni^2+^ 2p3/2, Ni^3+^ 2p_3/2_, Ni^2+^ 2p_1/2_, and Ni^3+^ 2p_1/2_, respectively [53]. Ni^3+^ is difficult to further oxidize due to its high valence. The observed 4 peaks at 706.7–723.8 eV correspond to Fe^2+^ 2p_3/2_, Fe^3+^ 2p_3/2_, Fe^2+^ 2p_1/2_, and Fe^3+^ 2p_1/2_ [54], respectively. In addition, the characteristic peak centered around 713.2 and 722.9 eV represent the satellite peak. Ni^2+^ and Fe^3+^, as the main valence states for NiFeLDH/ACBp, are beneficial to improving the OER performance [10,55]. Compared with NiFeLDH/A−CBp with AA–0 mmol, the XPS peak of Ni 2p and Fe 2p of that with AA–0.1 mmol transfer to higher binding energy, show stronger interaction between metals. The spectrum of O 1s is shown in Figure 4f; the peak at 530.19 eV is connected to the oxygen metallic bond, and the other peak at 531.09 eV is attributed to the oxygen vacancy or surface hydroxyl. The =O is related to the peak found at 532.49 eV [47,56]. Compared with the XPS spectra of AA–0 mmol and AA–0.1 mmol, the O 1s peak showed significant changes, with a smaller peak area of O at 532.49 eV and an increase in the area of oxygen vacancies or surface hydroxyl peaks at 531.09 eV, thanks to an increase in the number of hydroxyl groups in AA.

For water splitting, the OER catalytic performance of NiFeLDH/A–CBp samples with different AA contents was tested in 1 M KOH. Among them, AA–0.1 mmol exhibited the lowest overpotential (227 mV) at 10 mA·cm^−2^ (Figure 5a). It has excellent electrocatalytic performance compared to similar materials in the past (Appendix A) [57,58,59,60,61]. The introduction of AA can improve the Tafel slopes of catalysts, implying that AA can accelerate the charge transfer rate, as indicated in Figure 5b [42]. CV tests were carried out to further explore the electrochemically active area of NiFeLDH/A–CBp catalysts (Appendix A). The C_dl_ can be obtained by calculation from the curves of current density (CD) vs. scan rate in Figure 5c. The C_dl_ value of AA–0.1 mmol is the largest at 43.26 mF cm^−2^, indicating that more active sites of the catalyst can be exposed by adjusting the amount of AA. Hence, an NiFeLDH/A–CBp catalyst with 0.1 mmol AA has more active sites and a fast charge transfer rate, indicating better electrochemical catalytic properties (Figure 5d). As seen from Figure 5e, although the conductivity of the material is slightly weaker after the addition of AA, it does not affect the improvement of the catalytic performance. The charge transfer resistance of NiFeLDH/A–CBp is relatively small. Meanwhile, NiFeLDH/A–CBp exhibits a smaller series resistance and a higher Warburg impedance slope (Z_W_), suggesting increased conductivity to boost the electron transfer rate [62,63,64]. The voltage maintains 99.2% and the surface morphology and structure of NiFeLDH/A–CBp did not show significant changes after long-term stability testing, which exhibits the excellent ability of NiFeLDH/A–CBp.

Based on the good OER performance of the NiFeLDH/A–CBp catalyst, it firstly can be used as an anodic catalyst for water electrolysis, as shown in Figure 6. The cathode is made of a foam nickel molybdenum material [65]. The NiFeLDH/A–CBp catalyst shows high OER current density with a low overpotential of 227 mV at 10 mA cm^−2^ (Figure 6b). The performed stability test demonstrates excellent overall water solubility stability, which is the same as the change in the oxygen evolution OER stability test (Figure 6c) [66,67,68].

In order to reduce the electrolyzer voltage and lower the energy consumption of zinc hydrometallurgy, NiFeLDH/A–CBp was also employed as an anodic OER catalyst for zinc electrolysis. The effects of the current density and temperature on the current efficiency, electrolyzer voltage, and energy consumption were studied in an alkaline solution of 0.4 M ZnO and 6 M KOH.

Figure 7a shows the influence of the CD on three factors at 30 °C. With the increase of the CD from 300 to 2000 A·m^−2^, the current efficiency firstly decreases slightly, then rapidly due to the enhancement of hydrogen evolution at higher current densities [34]. The increased CD will increase the voltage drop caused by the electrolyte, electrode, and ohmic contact, increasing the electrolyzer voltage and power consumption in zinc electrolysis [69]. So, considering the electrolytic rate, a CD of 1000 A·m^−2^ is superior. Next, the influence of the electrolytic temperature was studied at 1000 A·m^−2^, as shown in Figure 7b. With the increase of the electrolytic temperature, the current efficiency first increased and then decreased. At 40 °C, the zinc electrodeposition efficiency reaches 99.38%, and the cathodic hydrogen evolution side reaction is greatly suppressed [70]. However, the electrolyzer voltage and energy consumption decreased first and then increased. Under high-electrolytic-temperature conditions, the dominant factor is the decrease of the hydrogen overvoltage, which is conducive to hydrogen evolution, thereby reducing the current efficiency and increasing the power consumption [34]. Furthermore, the temperature is too high, and the zinc powder is seriously dissolved. Hence, the optimal electrolytic conditions are at 40 °C with a CD of 1000 A·m^−2^ by comprehensive consideration.

Under the optimal experimental conditions, the electrolyzer voltage of foam nickel (NF) as anode is approximately 2.4 V, and the energy consumption is 2.056 kWh/Kg_Zn_. In contrast, a lower electrolyzer voltage of 2.08 V and energy consumption of 1.782 kWh/Kg_Zn_ can be obtained by NiFeLDH/A–CBp dropped onto NF (NF–CL) as an anodic catalyst (Figure 7c); these values are also significantly lower than that those reported, published, and shown in Figure 7e [34,35,37,39,71]), highlighting the excellent catalytic performance of NiFeLDH/A–CBp. In addition, NiFeLDH/A–CBp has good stability. The electrolyzer voltage with NF–CL is almost constant after electrolysis for 1 h, while it increases 2.92% with NF. The stability of NiFeLDH/A–CBp is further demonstrated by a long-time electrolytic test of 600 min (Figure 7d), and the voltage of NF–CL can be maintained at 99.88%, proving that an NiFeLDH/A–CBp catalyst can effectively reduce energy consumption in zinc electrolysis.

The cathodic products of electrodeposition zinc were also analyzed, as shown in Figure 8. The morphology of zinc powder obtained at different CDs for 1 h is characterized (Figure 8a–c). At a low current density of 500 A·m^−2^, dense, uniaxial, and nanoscale zinc powder is produced, which is easy to oxidize due to low ion mobility. With the increase of the CD to 1000 A·m^−2^, the deposition rate increased, promoting the growth of zinc nanoparticles. The zinc powder became flaky, and the size of the zinc powder increased to a micron level, which is closer to the requirement for commercial Zn powder. Therefore, the higher current density of 1000 A·m^−2^ is selected as optimal, though the electrolyzer voltage and energy consumption are a little higher than that they are at lower CD values. As the CD increases to 1500 A·m^−2^, the flaky zinc powder becomes thicker at a faster deposition rate, resulting in many particles gathering on the surface (Figure 8c) [72]. Figure 8d is the XRD diagram of the zinc products obtained at different temperatures. The characteristic peaks of all materials at different electrodeposition temperatures are well-matched with XRD standard cards of Zn, PDF#65-3358. The peaks at 36.29°, 38.99°, 43.22°, 54.32°, 70.08°, 70.63°, and 77.05° correspond to (002), (100), (101), (102), (103), (110), and (004) of Zn [72]. With the increase in temperature, the intensities of characteristic diffraction peaks gradually decrease, showing that the higher the temperature, the faster the deposition rate of cathode zinc ions and the worse the crystallinity of the products [72]. At 40 °C, the crystallinity of the obtained zinc powder is good, and the current efficiency is highest. Thus, 40 °C as the optimal electrolytic temperature is reasonable.

## 4. Conclusions

In summary, we obtained the conductive material CBp from waste tires and formed high-performance catalysts for water decomposition and zinc deposition through simple in situ synthesis. The SEM images reveal that the resulting NiFeLDH uniformly covers the surface of A–CBp. The optimized NiFeLDH/A–CBp catalyst demonstrated a higher C_dl_ of 43.26 mF·cm^−2^ and a lower overpotential (227 mV at 10 mA·cm^−2^), indicating increased active sites, electrochemical active area, and catalytic performance. In addition, NiFeLDH/A–CBp as an anodic catalyst presented a lower electrolytic voltage (2.08 V at 100 mA·cm^−2^), and the energy consumption (1.782 kWh/kg_Zn_) for Zn electrodeposition in an alkaline solution was much lower than that of 3.4 kWh/Kg_Zn_ in industrial acidic electrowinning. The proposed work provides an energy-saving OER catalyst derived from CBp for water splitting and Zinc electrowinning in industrial applications.

## Figures and Tables

**Figure 1 materials-16-03997-f001:**
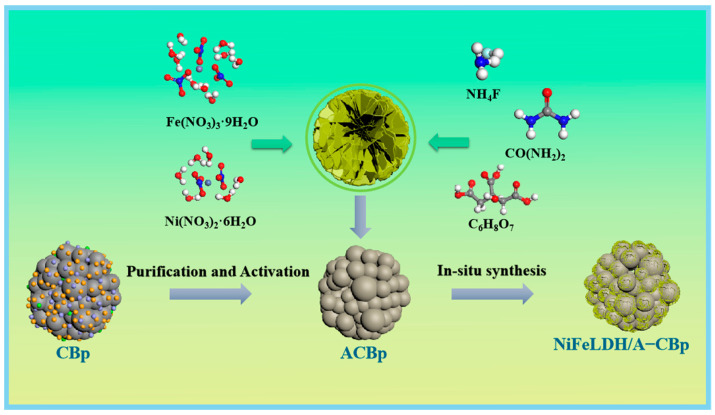
Schematic diagram of NiFeLDH/A–CBp preparation.

**Figure 2 materials-16-03997-f002:**
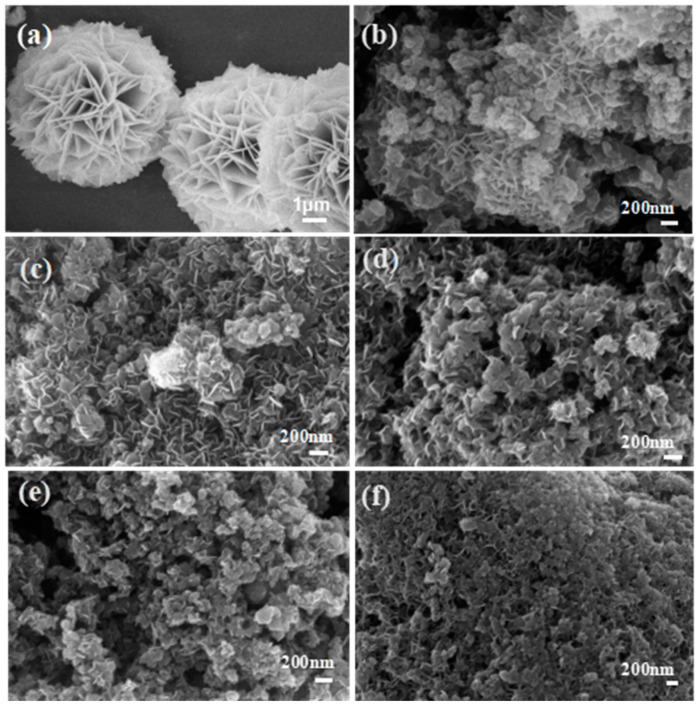
(**a**) SEM of NiFeLDH. SEM of NiFeLDH/A–CBp obtained by adding (**b**) 0 mmol, (**c**) 0.1 mmol, (**d**) 0.2 mmol, (**e**) 0.3 mmol, or (**f**) 0.4 mmol of AA.

**Figure 3 materials-16-03997-f003:**
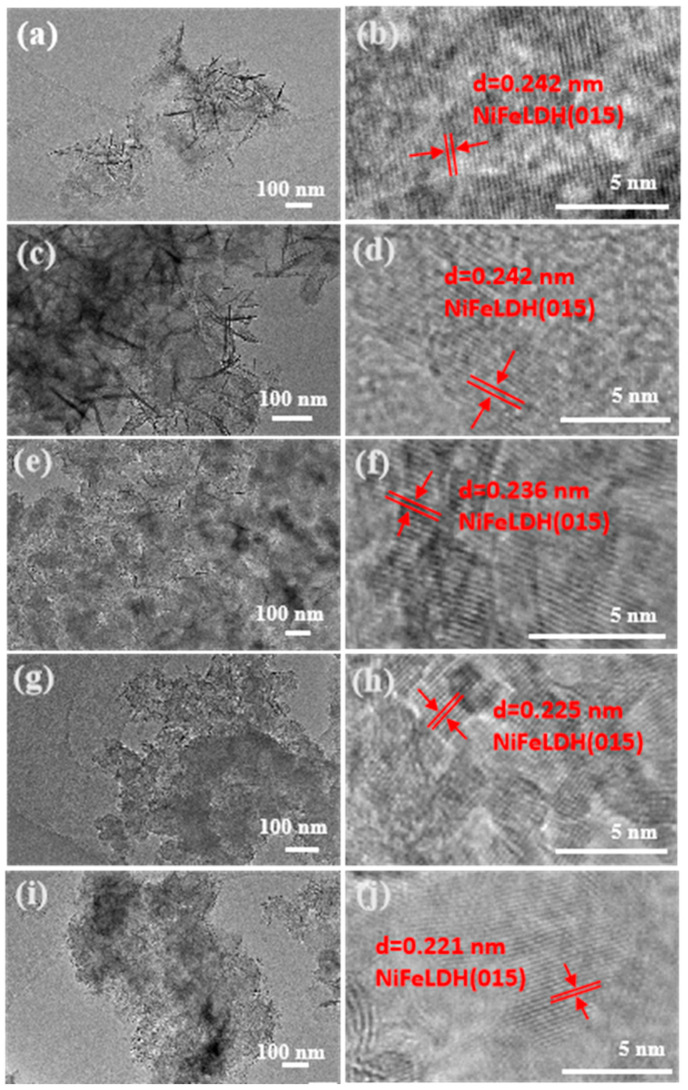
TEM of NiFeLDH/A–CBp obtained by adding (**a**) 0 mmol, (**c**) 0.1 mmol, (**e**) 0.2 mmol, (**g**) 0.3 mmol, or (**i**) 0.4 mmol of AA. (**b**,**d**,**f**,**h**,**j**) HRTEM of NiFeLDH/A–CBp corresponding to (**a**,**c**,**e**,**g**,**i**).

**Figure 4 materials-16-03997-f004:**
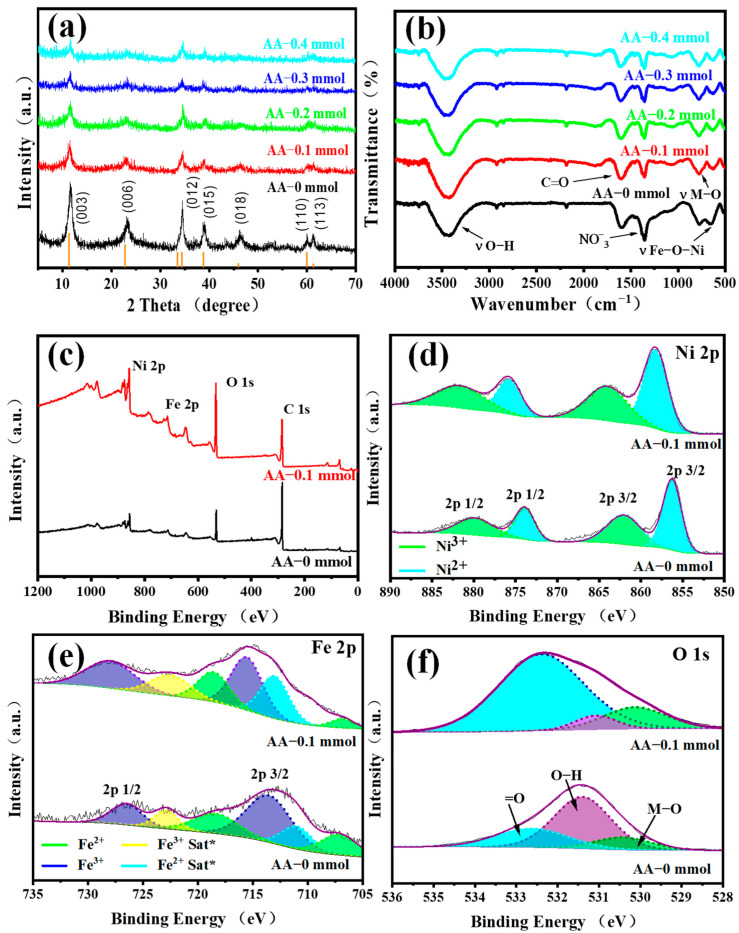
(**a**) XRD of NiFeLDH/A–CBp with different AA contents. (**b**) FTIR of NiFeLDH/A–CBp with different AA contents. XPS spectra of AA–0 mmol and AA–0.1 mmol, (**c**) XPS full spectrum, (**d**) Ni 2p spectrum, (**e**) Fe 2p spectrum and (**f**) O 1s spectrum.

**Figure 5 materials-16-03997-f005:**
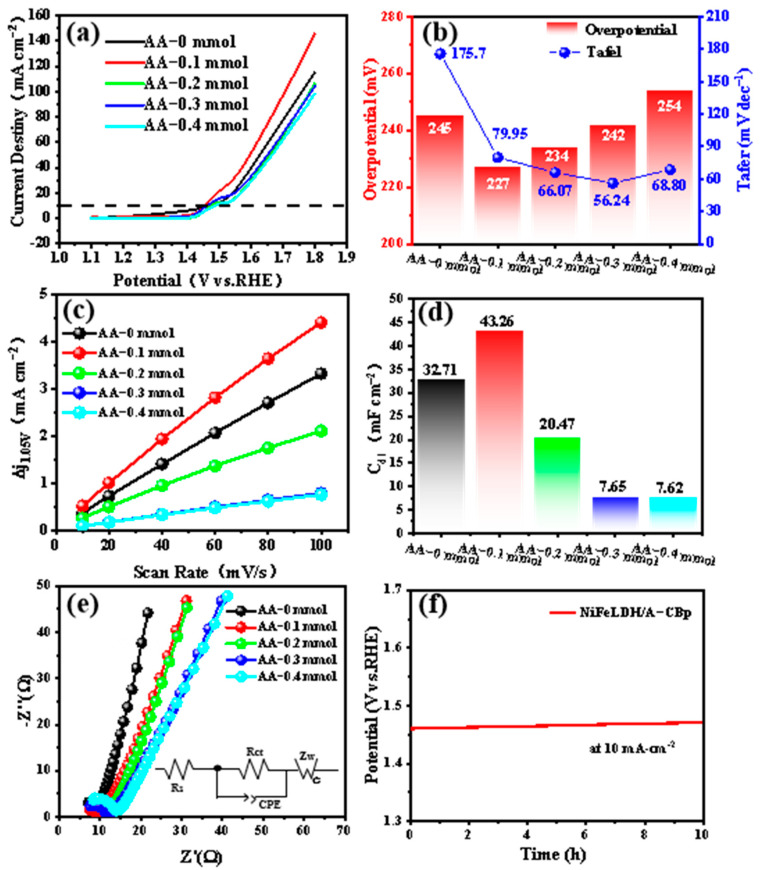
The electrochemical performances of NiFeLDH/A–CBp samples with different AA contents. (**a**) LSV curves, (**b**) Overpotential and Tafel curves, (**c**) C_dl_ curves, (**d**) C_dl_ values, (**e**) Nyquist plots (the illustration is the equivalent electrical circuit), and (**f**) CP of NiFeLDH/A–CBp catalyst obtained at 0.1 mmol AA content in 1 M KOH.

**Figure 6 materials-16-03997-f006:**
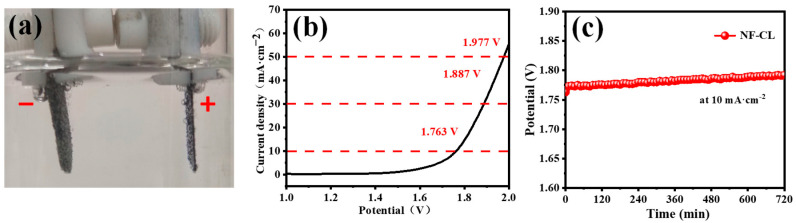
(**a**) The picture of total water hydrolysis test. (**b**) Polarization curve (LSV) of NF-CL by total hydrolysis test. (**c**) Stability test curve of NF-CL at 10 mA·cm^−2^ current density.

**Figure 7 materials-16-03997-f007:**
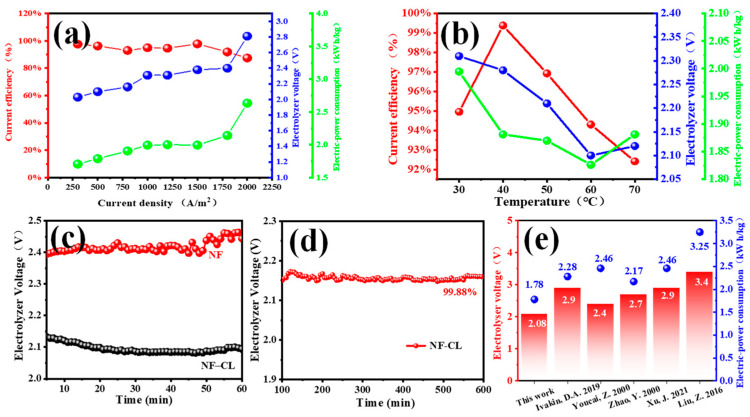
The curves of current efficiency, electrolyzer voltage and electric power consumption changed with (**a**) current density, and (**b**) electrolyte temperature with NiFeLDH/A–CBp loaded onto carbon paper. (**c**) Electrolyzer voltage curves of NF and NF–CL during electrolysis within 60 min. (**d**) Constant-current electrolysis curve of NF–CL in 0.4 M ZnO and 6 M KOH. (**e**) Comparison of electrolyzer voltage and electric power consumption reported in various works in the literature at 1000 A/m^2^ for alkaline zinc electrodeposition [34,35,36,37,38,39,71].

**Figure 8 materials-16-03997-f008:**
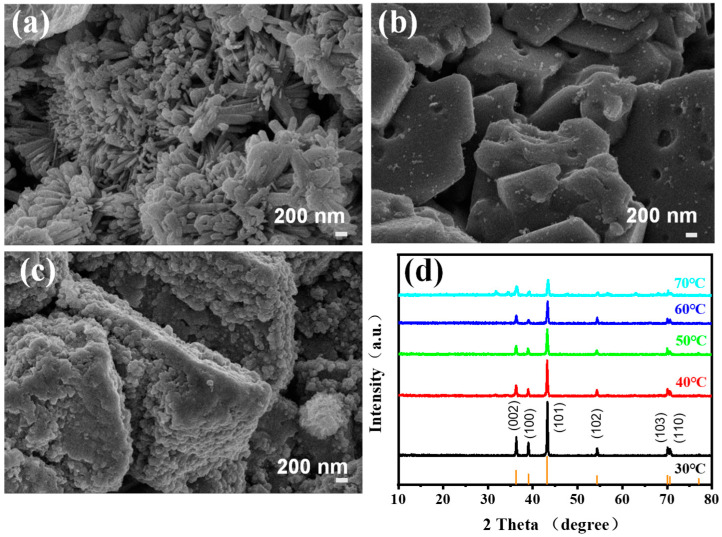
SEM images of produced Zn powder from 6 M KOH–0.4 M ZnO solution and at current densities of (**a**) 500 A·m^−2^, (**b**) 1000 A·m^−2^, and (**c**) 1500 A·m^−2^ at 30 °C. (**d**) XRD of cathodic products at different temperatures and 1000 A·m^−2^.

## Data Availability

The data are contained within the article.

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
