# Peer review of "In Situ Synthesis of NiFeLDH/A–CBp from Pyrolytic Carbon as High-Performance Oxygen Evolution Reaction Catalyst for Water Splitting and Zinc Hydrometallurgy"

_materials, 2023, doi:10.3390/ma16113997_

Round 1
Reviewer 1 Report
Comments for the paper
Manuscript Id: Materials 2356267
Manuscript Title: In situ synthesis of NiFeLDH/A−CBp from pyrolytic carbon as High-Performance Oxygen Evolution Reaction Catalyst for water splitting and zinc hydrometallurgy
The manuscript discusses the fabrication of NiFeLDH electrocatalytic materials for electrochemical water splitting and zing hydrometallurgy applications. This work is not innovative; however, preparing this method could be considered for publication after significant revisions. Therefore, some issues need to be addressed before its publication.
1. The abstract section must be revised to highlight the purpose of this work. General statements need to be removed. Must highlight achievements in detail.
2. The authors should give more exact details about the purpose of the work in the introduction section. What are the new insights? Paragraphs must be appropriately arranged.
3. SEM images must be discussed in detail. The picture quality is very poor. I don’t understand why the scale bar is not provided. Agglomerating of particles confirms the nanostructures? What are the authors trying to convey here?
4. All the figures standard need to be improved. It's hard to read the ligands and labels in the figures. Must be redrawn to show better representation to the readers.
5. XRD spectra- All the diffraction pattern must be labeled. Matching diffraction patterns must be specified.
6. XPS spectra- Figure 3b- All the survey peaks must be identified. What do you mean by Fe 2p3? XPS spectra must be compared with AA=0.0. O 1s spectra must be provided and discussed in detail.
7. In a schematic representation- Actization ?
8. Figure 5a. Electrode needs to be specified.
9. HRTEM images must be provided.
10. After OER durability test- Structural and surface features must be provided.
11. Line 118- 1-1.1 ? Voltage need to be specified.
12. All the SEM images- No scale bar given- no information provided. More attention must be paid to
13. Experiemental section. Were structures examined using FTIR? What structures?
14. In the results and discussion part, provide a comparison with previous similar materials Must tabulate the electrochemical performance with the literature.
15. Many space errors/punctuation errors must be solved.
16. Authors must discuss most parts of the results without bibliographic support! In many sections, references are needed. Please, by using similar papers that evidenced the same behavior, the Authors are encouraged to better describe the obtained results;
17. Conclusion section- must focus on future directions of these prepared materials?
18. There are many errors in the paper, so the Authors are encouraged to review the form and the English of the manuscript.
There are many errors in the paper, so the Authors are encouraged to review the form and the English of the manuscript.
Reviewer 2 Report
1. Please underscore the scientific value-added to your paper in your abstract. Your abstract should clearly state the essence of the problem you are addressing, what you did and what you found and recommend, that would help a prospective reader of the abstract to decide if they wish to read the entire article.
2. In your discussion section, please link your empirical results with a broader and deeper literature review.
3. The introduction part is too short. Increase its length with relevant supporting literature to develop the interest of readers to this article.
4. In materials section mention the CAS number and purity of the chemicals.
5. Mention the numbers with equations as equation number is missing in section 2.4.
6. In the Figure 2 and below figures, mention mmol quantities on the SEM images as well.
Reviewer 4 Report
Novelty of the Work:
A new data about, the иn situ synthesis of NiFeLDH/A−CBp from pyrolytic carbon as High-Performance Oxygen Evolution Reaction Catalyst for water splitting and zinc hydrometallurgy.
Significance of the Work:In this work, the purified and activated CBp is combined with nickel iron hydrotalcite (NiFeLDH) to form NiFeLDH/A−CBp catalyst for oxygen evolution reaction.
Introduction – Despite the extensive references, the authors have to do more relevant discussion.
Conclusion: The authors should give a more detailed discussion about the future application of the obtained materials.
Round 2
Reviewer 1 Report
Accept in present form
Minor editing of English language required
Reviewer 3 Report
Accept
Minor spell check required